# HSP90 Molecular Chaperones, Metabolic Rewiring, and Epigenetics: Impact on Tumor Progression and Perspective for Anticancer Therapy

**DOI:** 10.3390/cells8060532

**Published:** 2019-06-03

**Authors:** Valentina Condelli, Fabiana Crispo, Michele Pietrafesa, Giacomo Lettini, Danilo Swann Matassa, Franca Esposito, Matteo Landriscina, Francesca Maddalena

**Affiliations:** 1Laboratory of Pre-Clinical and Translational Research, IRCCS, Referral Cancer Center of Basilicata, 85028 Rionero in Vulture (PZ), Italy; valentina.condelli@crob.it (V.C.); fabiana.crispo@crob.it (F.C.); michele.pietrafesa@crob.it (M.P.); giacomo.lettini@crob.it (G.L.); francesca.maddalena@crob.it (F.M.); 2Department of Molecular Medicine and Medical Biotechnology, University of Naples Federico II, 80131 Naples, Italy; daniloswann.matassa@unina.it; 3Medical Oncology Unit, Department of Medical and Surgical Sciences, University of Foggia, 71100 Foggia, Italy

**Keywords:** HSP90, molecular chaperone, metabolism, epigenetics

## Abstract

Heat shock protein 90 (HSP90) molecular chaperones are a family of ubiquitous proteins participating in several cellular functions through the regulation of folding and/or assembly of large multiprotein complexes and client proteins. Thus, HSP90s chaperones are, directly or indirectly, master regulators of a variety of cellular processes, such as adaptation to stress, cell proliferation, motility, angiogenesis, and signal transduction. In recent years, it has been proposed that HSP90s play a crucial role in carcinogenesis as regulators of genotype-to-phenotype interplay. Indeed, HSP90 chaperones control metabolic rewiring, a hallmark of cancer cells, and influence the transcription of several of the key-genes responsible for tumorigenesis and cancer progression, through either direct binding to chromatin or through the quality control of transcription factors and epigenetic effectors. In this review, we will revise evidence suggesting how this interplay between epigenetics and metabolism may affect oncogenesis. We will examine the effect of metabolic rewiring on the accumulation of specific metabolites, and the changes in the availability of epigenetic co-factors and how this process can be controlled by HSP90 molecular chaperones. Understanding deeply the relationship between epigenetic and metabolism could disclose novel therapeutic scenarios that may lead to improvements in cancer treatment.

## 1. Introduction

Molecular chaperones belonging to the heat shock proteins (HSPs) family play a fundamental role in many cellular processes, ensuring normal cellular functions, and their overexpression in many cancer types favors cancer cell adaptation to various stress conditions in the tumor microenvironment. HSP90 chaperones, in particular, regulate, in a direct or indirect manner, many cellular processes such as signal transduction, cell proliferation, angiogenesis, and bioenergetics.

As it is known, tumor cells differ from normal cells, for their metabolic profile and metabolic rewiring is currently considered a hallmark of cancer. Cancer metabolic plasticity allows tumor cells to survive in unfavorable environments and to maintain their homeostasis. Metabolic rewiring is determined by a combination of genetic events, for example mutations, and non-genetic factors, such as adaptation to the tumor microenvironment and nutrient availability [1]. The major biochemical pathways affected by cancer reprogramming are glycolysis and tricarboxylic acid (TCA) cycle. Indeed, cancer cells need a continuous production of adenosine triphosphate (ATP) in order to satisfy their energetic requirements and to sustain their rapid growth and proliferation. Moreover, highly proliferative cells, like tumor cells, require the active biosynthesis of macromolecules, such as lipids, nucleotides, and amino acids, in order to support cell replication, and to preserve redox homeostasis in order to balance the increased production of toxic reactive oxygen species (ROS). Even if ATP production via oxidative phosphorylation (OXPHOS) is 18-fold more efficient than glycolysis [2], the majority of cancer cells prefer glycolysis as an ATP source, because it represents a metabolic solution that satisfies the sum of all of the tumor demands. At the same time, the TCA cycle is also useful for cancer cells, because its intermediates are precursors of lipids and nucleotides. For this reason, cancer cells activate anaplerotic pathways, such as glutaminolysis, to sustain the TCA cycle in the production of intermediate metabolites [1]. Thus, although cell proliferation is predominantly sustained by glucose and glutamine catabolism, a unique metabolic program cannot be defined to describe the metabolic changes for all tumors. Each cancer type exhibits characteristic metabolic features that are a mirror of the genetic, epigenetic, and microenvironmental heterogeneity.

Epigenetics generally refers to the chemical modification of DNA and histones, affecting the accessibility of DNA to transcription factors and eliciting changes in the gene expression profile. Epigenetic mechanisms remodel the chromatin structure through both DNA methylation (CpG methylation) and histone post-translational modifications, such as acetylation, methylation, ubiquitination, SUMOylation, phosphorylation, acylation, and O-linked N-acetylglucosamine modification (O-GlcNAcylation) [3]. Enzymes, noted as writers, readers, and erasers, are responsible for chromatin remodeling. The first epigenetic players introduce the chemical tags on chromatin; the readers identify and decipher those modifications, attracting locally transcription factors; and the erasers remove these chemical modifications, reversing the chromatin state shaped by the writers [4].

The strict regulation of the epigenetic processes is fundamental for the health and correct development of organisms. An altered pattern of epigenetic modifications, especially methylation and acetylation, is central to the onset of many common human diseases, including cancer, because of the potential silencing of tumor suppressor genes and/or the activation of oncogenes with inauspicious results. In the context of human cancer, emerging evidence suggests a reciprocal crosstalk between epigenetic mechanisms and metabolic changes. Indeed, the metabolic rewiring of cancer cells requires epigenetic modifications so as to achieve the best and most rapid response to adapt to the adverse conditions that rapidly fluctuate in the tumor microenvironment. Conversely, the tumor metabolism provides molecular substrates that control and sustain epigenetic mechanisms, and this is relevant for tumor progression [5].

Recent evidence suggests a role for HSP90 as and “epigenetic capacitor” for phenotypic variation. HSP90 with its chaperoning activity contributes to the phenotypic plasticity [6] and modulates epigenetics by interacting with chromatin, chromatin regulators, and epigenetic effectors. As HSP90 may promote phenotypic diversity favoring metabolic reprogramming with significant gene expression remodeling, in this review, we discuss the role of HSP90 chaperones in the complex interplay between metabolism and epigenetics. Considering that the heat shock system is activated in many cancer types, resulting in poor prognosis, and HSP90 may be considered a hub of the cross-interaction between metabolism and epigenetics, the metabolism–HSP90–epigenetics network highlighted in this review may represent a novel therapeutic target for successful cancer treatment.

## 2. Heat Shock Proteins

Molecular chaperones, called also HSPs, were discovered and characterized as molecule whose expression is massively induced in cells exposed to stress conditions (thermal shock, environmental and chemical factors, and pathological changes) [7,8]. Physiologically, HSPs are the main class of molecular chaperones responsible for (i) coordinating the correct folding of partially folded and/or denatured proteins, (ii) controlling the functional conformation of proteins, and (iii) preventing the irreversible formation of damaged protein aggregates [8,9].

HSPs include multiple families, whose members originate from an ancestral gene through a gene duplication mechanism, and are characterized by a high sequence and structural homology [10]. Based on the molecular weight, sequence, and structural and functional homology, six families of HSPs have been identified, namely: HSP20, HSP40, HSP60, HSP70, HSP90, and HSP100 [9,11]. Each family participates in the folding of the proteome, displaying subtle dissimilarities in the properties and functions of the substrate proteins or client proteins, and allows for proteins to take the functional active tertiary structure [9,12]. Performing chaperoning activities, HSPs are the main protagonists in maintaining protein cell homeostasis [13], and they play a key role in the fundamental processes for cell survival, such as (i) protein synthesis modulation, (ii) the regulation of cell signaling translation pathways, and (iii) RNA processing [14,15,16]. These characteristics make HSPs so essential in the biology/physiology of normal cells that dysfunction in their synthesis or activity causes numerous pathological conditions, including neurodegenerative diseases, cardiovascular illnesses, and cancer [17,18]. In particular, during malignant transformation and tumor progression, highly proliferating tumor cells synthetize many mutated and/or aberrant proteins that need to be folded [19]. As a consequence, they upregulate HSPs in order to assure the accurate resolution of stress folding and respond to various stress conditions in tumor microenvironment, such as metabolic stress, hypoxia, nutrient deficiency, and drug therapy [17,20]. The over-expression of HSPs has been observed in multiple types of cancers, such as prostate, cervical, ovarian, renal, brain, lung, colorectal, hepatocellular, breast carcinomas, and myeloid leukemia [17]. Besides the effect of chaperone overexpression during tumorigenesis, it was also proposed that cancer cells frequently present HSPs associated to heteroprotein complexes with client proteins and co-chaperones, whereas normal cells display HSPs in a homodimeric state [21].

Numerous studies sought to better understand which members of different HSPs families cooperate in controlling the hyper-activated signaling networks of cancer cells in order to make anticancer therapies more effective. These efforts led to the characterization of the main functions controlled by HSPs families [2,9]. Recently, it has emerged that HSPs can be released from cells and remain on the external cell surface. These extracellular HSPs (eHSPs) are involved in the activation of intracellular signaling, as well as in the intercellular communication and inflammatory and immune processes [22,23].

### HSP90 Family

Members of the HSP90 family (molecular weight of about 90kDa) are highly conserved molecules that promote the folding of neo-synthesized or incorrectly folded proteins, and block their aggregation [24]. The structural, biochemical, and molecular characteristics of HSP90 have been widely revised [24,25,26]. The HSP90 family includes the following: (i) two HSP90 isoforms called HSP90α and HSP90β, localized in the cytosol and in the nucleus [27]; (ii) GRP94, localized in the endoplasmic reticulum [28] (ER); and (iii) TRAP1, mainly localized in the mitochondria, but also in ER [29]. Structurally, family members are constituted by a N-terminal domain that binds ATP, a C-terminal domain that contains subcellular localization sequences, and a middle domain, essential for interactions with client proteins and for functional and structural flexibility [24,30]. The two isoforms, HSP90α and HSP90β, originated by gene duplication, are localized in the cytoplasm, and have an 85% sequence homology. HSP90α is the inducible form, while HSP90β is the constitutively expressed form. Both form dimers, and their dimerization is essential for the functionality of HSP90 in physiological conditions. Because of the remarkable structural and functional similarity, the majority of studies did not provide a clear distinction between the two isoforms [30].

Similarly to other molecular chaperones, the main function of the HSP90 isoforms is to assist protein folding, prevent aggregation, and attend to the refolding of denatured proteins in cooperation with other co-chaperones [31]. Approximately two hundred HSP90 client proteins have been described, and, among these, there are proteins involved in several cell functions, some of them extremely relevant in human carcinogenesis [32]. The critical function controlled by the HSP90 chaperones are (i) the cellular signaling pathways [27], (ii) cell survival [33], (iii) cytoskeletal integrity [34], (v) cell cycle [27], and (vi) cellular differentiation [35]. Finally, the extracellular form of HSP90α (eHSP90α) is involved in immunological functions, cell motility, and wound healing [23,36].

GRP94 differs from other components of the HSP90 family, because of its localization exclusively in the ER and its functional selectivity [28]. In fact, GRP94 performs quality control, promoting the folding and assembly of a small number of substrate proteins (i.e., secreted and membrane proteins). These include the transferrin receptor and MHC class 1 proteins, specific domains of immunoglobulin light chains, toll-like receptors, integrins, thyroglobulin, and insulin-like growth factor [28]. Furthermore, GRP94 is able to bind Ca^2+^, and, together with other ER resident chaperones, is involved in Ca^2+^ homeostasis. Thanks to this specific function, GRP94 is essential for cell survival [37,38].

TRAP1, initially identified as a molecular chaperone of tumor necrosis factor receptor 1 (TNFR1) [39] and retinoblastoma protein (Rb), is the component of the HSP90 family located in the mitochondria [40,41]. However, recently, it has been shown that TRAP1 also resides in the ER, where it orchestrates the biogenesis and quality control of nascent polypeptides by interacting with several translation factors (eIF4A, eEF1A, and eEF1G) and the proteasome regulatory particle TBP7/Rpt3 [42,43]. Thus, it has been proposed that TRAP1 regulates the folding of its client proteins by direct protein interaction, mainly in the mitochondria [44], or by a co-translational quality control on the ER, preventing their proteasomal degradation [43]. Among the TRAP1 client proteins, there are key regulators of cell survival, such as cyclophilin D (CypD), a mitochondrial protein that maintains mitochondrial integrity by preserving cells from apoptosis [45], and the mitochondrial isoform of Sorcin, a Ca^2+^-binding protein responsible of Ca^2+^ homeostasis [46]. The TRAP1 client protein network also includes the proteins responsible for cell cycle progression, such as cyclin-dependent kinase 1 (CDK1) and mitotic arrest deficient 2 (MAD2) involved in the G2-M progression and the formation of the mitotic spindle [47]. Recently, the proteins responsible for intracellular signaling were also described as being controlled by the TRAP1 pathway, namely: (i) β-catenin, which regulates Wnt signaling [48], and (ii) Raf murine sarcoma viral oncogene homolog B1 (BRAF), a serine/threonine kinase responsible for cell growth through the hyper-activation of mitogen-activated protein kinase 1/extracellular signal-regulated kinase 1/2 (MAPK/ERK1/2) signaling [49]. Finally, succinate dehydrogenase (SDH) and cytochrome oxidase, components of the respiratory chain complexes that are responsible for energy metabolism, have been described as TRAP1 client proteins [50].

## 3. Role of HSP90 Family Members in Cancer Hallmarks

In cancer cells, single genetic mutations in elements of regulatory mechanisms cause the aberrant activation of normal physiological processes, and are responsible for cell transformation and tumor progression [51]. These irregular biological characteristics, common to most tumors and defined as cancer hallmarks, are acquired during the multistep development of human cancers [52]. These hallmarks are classified in six groups, namely: (i) independence from extracellular growth factors/signals, (ii) insensitivity to growth inhibitory mechanisms, (iii) evasion from programmed death programs, (iv) unlimited replication potential, (v) activation of angiogenesis, and (vi) a high capacity to invade surrounding tissues and form distant metastases. Moreover, two emerging characters have been identified that delineate tumor traits, such as the ability to escape from immune response and to rewire energetic metabolism. According to Hannan et al., the principal characteristics enabling the acquisition of both core and emerging hallmarks are tumor promoting inflammation, and genome instability and mutability [52].

Many of the proteins involved in the molecular mechanisms underlying tumor hallmarks are client proteins of the HSP90 family members. The HSP90 chaperone’s role in controlling the core cancer hallmarks, in blocking the immune system, and in promoting inflammation has been widely reviewed [53,54,55]. Thus, this review will focus on the role of the HSP90 family members in metabolic and genome instability/epigenetic reprogramming that represent emerging and enabling cancer hallmarks.

### Role of HSP90 Family Members in Cancer Metabolic Reprogramming

Metabolic rewiring is a distinctive feature of cancer cells characterized by a high proliferation rate. Indeed, proliferating tumor cells require both ATP and high amounts of substrates for the synthesis of biological macromolecules, such as nucleic acids, proteins, and lipids, which are necessary in order to sustain and maintain fast-growing tumor cell biomasses in a stressful microenvironment [56]. Biological macromolecules are produced from metabolic intermediates of the glycolytic pathway and the citric acid cycle [57]. Therefore, in cancer cells, the most important metabolic change is the increased glucose consumption in parallel with the inhibition of pyruvate oxidation and its conversion into lactate in the presence of oxygen [58]. The prevalence of glycolytic metabolism, even in the presence of oxygen, is called the Warburg effect [59].

On the other hand, the regulation of the redox state (ratio NAD^+^/NADH and NADP^+^/NADPH), is another requirement that proliferating tumor cells must satisfy [60]. This leads to changes in the glucose metabolism, with deviation toward biosynthetic pathways, as the pentose phosphate pathway that indirectly cooperate with mitochondrial TCA cycle allows for the improved biosynthesis and generation of reducing equivalents from TCA cycle intermediates [60]. Moreover, in some tumors, the mitochondria remain fully functional, and oxidative phosphorylation (OXPHOS) continues to be the main source of ATP and an important driver of tumor progression. Indeed, mitochondrial respiration can (i) contribute to oncogenes-dependent transformation, (ii) support high-demand energy mechanisms such as protein translation, (iii) promote drug-resistance, and (iv) regulate cell motility and metastasis formation [61]. Intriguingly, cancer cells are characterized by a metabolic plasticity, and are able to shift from one metabolic pathway to another, so as to adapt to the environmental conditions [62].

Members of the HSP90 family play an imperative role in regulating the subtle equilibrium between glycolytic metabolism and OXPHOS, in response to the changes in the tumor microenvironment, such as nutrient and oxygen deprivation and exposure to toxic agents, triggering cell transformation and tumor progression [63,64,65]. HSP90 chaperones can regulate metabolic rewiring, either indirectly by altering the HSP90-dependent signaling pathways that control the expression of the proteins involved in complex metabolic networks, or directly by controlling the stability, conformation, and functional activity of some metabolic enzymes.

Numerous reports have shown that HSP90 indirectly regulates metabolic pathways through oncogenic signaling. In such a context, several authors described HSP90 upregulation in human malignancies as a key event in carcinogenesis and tumor progression [66]. Indeed, HSP90 interacts with and modulates several of the cell signaling pathways involved in metabolic plasticity: (i) c-Myc, which regulates the expression of nutrient transporters on the plasma membrane and glycolytic enzymes [67,68]; ii) hypoxia-inducible factor 1 alpha (HIF1α), which triggers the shift from OXPHOS to glycolysis in hypoxic conditions, inducing the expression of the genes encoding for glucose transporters and glycolytic enzymes, and activating the transcription of genes decreasing OXPHOS [69,70]; (iii) AKT/PKB, an effector of 3-kinase phosphoinositide (PI3K) involved in cell proliferation, metabolism, and antiapoptotic responses [71,72]; (iv) ErbB2, which promotes glycolysis [73,74]; and (v) cSRC, which increases glycolysis, and, after its translocation in mitochondria, downregulates the activity of the respiratory chain complexes and ATP production [63,75].

Extensive literature has shown that HSP90 can influence cancer metabolism by directly binding glycolytic enzymes. In 2002, Nakamura et al. demonstrated a direct interaction between HSP90 and glyceraldehyde 3-phosphate dehydrogenase (GAPDH) [76], while Xu et al. showed that HSP90 binds directly to pyruvate kinase muscle isozyme 2 (PKM2), increasing its stability and reducing its proteasome-dependent degradation [64]. Mechanistically, HSP90 and PKM2 form a complex with glycogen synthase kinase-3β (GSK3), which phosphorylates PKM2 at Thr-328, and the phosphorylation of PKM2 is essential for protein stability and biological functionality. Of note, the expression of the two proteins correlates in hepatocellular carcinoma HCC tissues [64]. However, despite several years of research, the exact mechanism by which HSP90 chaperones regulates cancer metabolism is still a matter of investigation, and the results are sometimes conflicting. In fact, some studies showed that HSP90 does not influence glucose uptake [77], but it seems to indirectly promote glycolysis through direct interaction with Tom40, one of the components of the mega mitochondrial transport TOM complex. Indeed, the inhibition of HSP90 causes the attenuation of glycolysis and a metabolic shift toward OXPHOS, and an increased production of succinate, an intermediate of the Krebs cycle. This occurs through a reduction of the incorporation of Tom40 and the incorrect assembly of the mitochondrial TOM complex, a consequent reduction of the import of the voltage dependent anion channels (VDACs) in the mitochondrial membrane and a detachment of the first key enzyme of glycolysis, hexokinase II (HKII), bound to VDAC [78]. It is known that HKII, when bound to VDAC, is more active, as the enzyme can easily find the ATP indispensable for its enzymatic activity. The inhibition of HSP90 causes the detachment of HKII from VDAC and the consequent inhibition of glycolytic pathway [78].

In the last years, the role of the HSP90 mitochondrial isoform, TRAP1, has increasingly emerged [65,79]. In particular, TRAP1 seems to modulate tumor energy metabolism by activating glycolysis and repressing OXPHOS, and this function is context- and tumor-specific [65]. Interestingly, TRAP1 upregulation has been described in several human malignancies, with a prevalent glycolytic metabolism (i.e., colon, breast, lung, prostate, nasopharyngeal, and thyroid carcinomas) [65], and a proteomic analysis of the TRAP1 client protein network demonstrated that TRAP1 is co-expressed with most of its interactors in human colorectal carcinomas, and that the overexpression of TRAP1 and six of its client proteins is predictive of poor outcomes [80]. Conversely, selective human malignancies (i.e., kidney, ovarian, and cervical carcinomas) are characterized by a prevalent oxidative metabolism and a consequent TRAP1 downregulation [65]. Chae et al. showed that the inhibition of TRAP1 in the mitochondria by Gamitrinib causes the release of HKII from the outer mitochondrial membrane, blocking its enzymatic activity and causing a reduction of glucose consumption, lactate production, and ATP generation in prostate cancer [81] (Figure 1). On the other hand, TRAP1 can interact with specific respiratory chain complexes, contributing to their stability/activity [65]. Indeed, TRAP1 binds the respiratory chain complex II (i.e., succinate dehydrogenase—SDH) and IV (i.e., cytochrome oxidase), and inhibits the activity of SDH with a consequent reduction of oxygen consumption. Interestingly, this mechanism favors the accumulation of succinate, which stabilizes HIF1α by blocking the prolyl hydrolases responsible for its ubiquitination-dependent degradation. Therefore, when the protein levels of TRAP1 are elevated, glycolysis is the main source of energy for cancer cells, and HIF1a stabilization represents a major mechanism that links TRAP1 to tumorigenesis [50], thus suggesting an oncogenic role for this molecular chaperone.

Conversely, Yoshida et al. suggested that TRAP1 might act as an oncosuppressor, with its expression being downregulated in specific tumors. These authors showed that high protein levels of TRAP1 decrease mitochondrial respiration and ATP production, while cell lines with low expressions of TRAP1 are characterized by a higher activity of complex IV and high ATP levels, and use oxidative phosphorylation and not glycolysis as the main energy source. Furthermore, intermediates of the citric acid cycle and anaplerotic substrates, such as those resulting from the oxidation of fatty acids and the NAD^+^/NADH ratio, are higher in low TRAP1 background compared with the cell lines with a high TRAP1 expression. Finally, they demonstrate that TRAP1 regulates mitochondrial respiration by binding and inhibiting the functions of the oncogene cSRC, and acting as tumor suppressor [63]. 

More recently, Matassa et al. demonstrated that TRAP1 is downregulated in ovarian cancer, and that this is responsible for platinum resistance. Indeed, TRAP1 silencing increases mitochondrial respiration without affecting glycolysis in ovarian cancer [82], and the isogenic cell lines of cisplatin-resistant ovarian carcinoma derived from patients after chemotherapy showed lower levels of TRAP1, increased oxygen consumption, and reduced glycolysis compared with the sensitive cell lines derived from the same patients before chemotherapy. These results indicate a close correlation between the TRAP1 expression, oxidative metabolism, and drug resistance [82].

Altogether, these studies confirm the hypothesis that cancer cells modulate the TRAP1 expression to remodel the balance between the glycolytic and oxidative metabolism in response to extracellular stimuli, and that this may be relevant for drug resistance and tumor progression. However, further studies are still ongoing in order to address whether the inhibition of the HSP90 chaperone may represent an anticancer strategy to target cancer cell metabolism and revert drug resistance in selective tumor types [83].

## 4. HSP90 as an ‘Epigenetic Capacitor’ for Phenotypic Variation

In 1942, C.H. Waddington in his “Nature” letter introduced the term “canalization” to describe the stable, normal course of development of living organisms [84]. He hypothesized that, during evolution, specific environmental or genetic factors can introduce novel adaptive characters, and the resulting novel phenotype may be fixed in a population (process called “assimilation”) [85], leading to advantageous phenotypic variants [86]. Approximately fifty years later, in 1998, Rutherford and Lindquist assumed that stress conditions, such as heat shock, unmask hidden phenotypic variations, and postulated that HSP90 might act as a “genetic capacitor” [87]. In *Drosophila melanogaster*, in fact, they demonstrated that the loss of HSP90 activity, by mutations or pharmacological inhibition with geldanamycin, showed previously masked phenotypic variations [88]. Even though Rutherford and Lindquist did not identify the targets of HSP90 in either *Drosophila melanogaster* [87] or in *Arabidopsis thaliana* [89], likely due to the high number of HSP90 client proteins, most of which involved in signal transduction [90], Ruden et al. proposed that not only genetic variations, but also epigenetic modifications of the chromatin state are responsible for these phenotypic variations. Interestingly, HSP90 may act not only as “genetic capacitor”, but also as an “epigenetic capacitor” for phenotypic variations [87]. They coined the term “epigenetically sensitized” to refer to “a chromatin modification that does not yet induce a new morphological phenotype, but it is on the verge of producing a new morphological phenotype” [91]. Sollars et al., in fact, by using an isogenic strain of *Drosophila melanogaster*, demonstrated that mutations in HSP90 or its pharmacological inhibition reveal an alteration in the chromatin state, itself responsible for the phenotypic variation of this strain [91]. Thus, “Epigenetic assimilation” persists, even in the absence of genetic variation and is epigenetically heritable. Therefore, the conclusion of the authors is that for the selection and fixation of the new phenotypic variation of the “genetic variation” can unnecessary. Altogether, this evidence suggests that HSP90 chaperones are likely involved in the controlling cell phenotypes through epigenetic modifications.

### 4.1. Epigenetic Regulation of Gene Expression in Cancer Cells

The most representative epigenetic modifications, frequently aberrant in cancer cells, are acetylation and methylation. Epigenetic methylation consists in the transfer of one methyl group from S-adenosyl-L-methionine (SAM), the universal methyl donor in the cell, to DNA or histone protein tails. DNA methyltransferases (DNMTs) are writers of DNA, because they catalyze the methylation of the 5′-position of cytosine, and this is associated with gene silencing. The DNA erasers, known as ten eleven translocation hydroxylases (TETs), remove the methylation marks on cytosine residues. Histone methylation can occur on lysine (K) or arginine (R) residues of both H3 and H4 histones. One single lysine residue can be methylated once, twice, or three times by histone lysine methyltransferase (KMTs). Protein arginine methylatransferases (PRMTs) are the enzymes responsible for the transfer of methyl groups to arginines. The erasers of methylation marks on lysine residues are divided into two families, because of their distinct enzymatic mechanisms and structures. The lysine specific demethylases (LSD1 and LSD2) are flavin adenine dinucleotide (FAD)-dependent demethylases specific for removing lysine mono- and di-methyl marks. The JmjC family (JHDMs) is a large group of enzymes, capable of recognizing and acting on all three methylation states of lysine.

The ε-amino group acetylation of lysine residues is an important histone modification and consists in the transfer of the acetyl group from acetyl-CoA to lysine residues on histones, leading to the formation of an open chromatin structure, associated with active transcription. The reaction is catalyzed by two families of histone acetyltransferases (HATs), namely: (i) Type A, which are localized in the nucleus and operate on chromatin-associated histones, and (ii) Type B, which are cytoplasmic enzymes involved in post-translational modifications of newly translated histones. The erasers of histone acetylation are known by the acronym HDACs, and remove the ε-amino acetyl group from lysine residues on histones, resulting in a compact transcriptionally repressive chromatin organization [4] (Figure 1).

### 4.2. HSP90 Epigenetic Mechanism of Action

Cells react to intra- and extra-cellular stimuli through the reprogramming of the gene expression profile. These changes may be transient or may become inherited by epigenetic mechanisms, maintaining the novel gene expression signature even if the original signal does not persist [92]. The elucidation of this phenomenon, called transgenerational epigenetic inheritance (TEI), represents a challenge for researchers, especially those involved in the study of diseases with an environmental influence, such as cancer [93].

HSP90 is a “stress sensor” facilitating client folding and stability, therefore with its activity contributes to phenotypic plasticity [6]. Environmental stresses and changes in cell physiology modulate HSP90 activity and its interactions with specific client proteins. As a consequence of this perturbation, HSP90 interaction with its client proteins is a dynamic process, and extreme phenotypes can be selected.

Historically, the majority of HSP90 client proteins are cytosolic [94]. However, differently from other chaperones, HSP90 clients are not an unspecific sampling of all proteins, but a well-defined set of proteins, enriched in the transcriptional factors and kinases [95] involved in signal-transduction pathways, chromatin remodeling, cell cycle regulation, and protein trafficking [25]. Moreover, a small percentage of HSP90 can be found in the nucleus (2%–3% in *Drosophila melanogaster*) [96,97], and many studies have revealed the physical interaction of HSP90 with a cohort of nuclear proteins implicated in gene transcription and epigenetic regulation [90]. This nuclear activity of HSP90 represents the prerequisite to understand the HSP90 epigenetic mechanism of action [98].

Recent evidence from the literature indicates that HSP90 can epigenetically regulate gene expression in a direct or indirect manner, as follows: (i) HSP90 is a chromatin-bound protein, and (ii) HSP90 binds to several chromatin regulators or epigenetic effectors [90] (Figure 2).

ChIP-seq experiments revealed that HSP90 directly interacts with chromatin-bound forms of nuclear clients and frequently this occurs near the transcriptional starting site of coding genes [90]. Very little information is available about the mechanism of HSP90 translocation in the nucleus. While it has been suggested that HSP90 phosphorylation is the prerequisite for its nuclear translocation in Drosophyla [96], this issue is still a matter of investigation in mammalian cells. According to the model proposed by Sawarkar et al. in *Drosophila melanogaster*, HSP90 localizes near the promoter of both the coding and noncoding genes, including many microRNAs (miRNAs) [99]. Nevertheless, gene activation needs significant alterations in the chromatin structure. In support of this thesis, early experiments in *Drosophila melanogaster* reported a HSP90 interaction with the chromatin domains involved in the active gene transcription [100]. Therefore, HSP90 is a chromatin-remodeling regulator, influenced by environmental changes, and it is able to switch the chromatin from a permissive state to a non-permissive state for transcription.

Secondly, the interaction between HSP90 and the chromatin may be indirect, as HSP90 interacts with and regulates several chromatin regulators or epigenetic effectors. For instance, HSP90 controls RNA polymerase II pausing, and this occurs by stabilizing the negative elongation factor complex (NELF), as demonstrated by the computational and biochemical analyses [6]. Moreover, a connection between the HSP90 and chromatin regulator factors has been proposed. According to this model, among the HSP90 client proteins, two novel HSP90 co-chaperones were identified in an integrated proteomic and genomic study in yeast [101], as follows: Tah1p (TPR-containing protein associated with HSP90) and Pih1p (protein interacting with HSP90), which link HSP90 to the chromatin remodeling factor Rvb1p (RuvB-like protein 1)/Rvb2p. This observation suggests a relationship between HSP90 and the epigenetic regulation mechanisms [93].

Another mechanism was proposed to explain the “capacitor” function of HSP90 in the morphological and phenotypic evolution [93], regarding a supposed role of HSP90 in the regulation of the Polycomb Group (PcG) and Trithorax Group (TrxG). Within the plethora of chromatin regulators, PcG and TrxG are among the most ancient and evolutionarily conserved chromatin regulators [90]. PcG and TrxG are catalytic elements of the epigenetic complexes regulating cell-lineage specification during normal growth with opposite roles, as follows: PcG represses and TrxG activates the developmental genes [97,102,103]. PcG proteins maintain “repressive chromatin marks” on the histone 3 lysine 27 tri-methylation (H3K27me3), TrxG proteins, instead, induce “active chromatin marks” on the histone 3 lysine 4 tri-methylation (H3K4me3) by Trithorax and Ash1, two client proteins of HSP90. Therefore, stress-induced inactivation of HSP90 and its pharmacological inhibition cause a switch from active to repressed chromatin, because of the degradation of Trithorax, with consequent gene expression downregulation.

Drosophila Trx is a member of the suppressor of variegation 1 (SET1; enhancer of Zeste and Trithorax) domain family of H3K4 methyltransferases, and its human orthologous is mixed lineage leukemia protein-1 (MLL1) [97,104]. Among the human SET-related family members, MLL1 plays a fundamental role in cell growth and hematopoiesis, and is involved in myeloid and lymphoblastic leukemia [105], as well as in solid tumors [106,107]. MLL1 is an HSP90 client protein itself, and rising studies showed that HSP90 regulates MLL family members by interacting with epigenetic regulators, including SMYD2 and SMID3, two components of the SET domain-including histone methyltransferases [108]. With regard to cancer, SMID3 has been suggested to play a role in the regulation of HSP90-mediated estrogen receptor (ER), with implications in uterine development and cancer [87]. Equally, PcG homologs are conserved in human species, where the polycomb-repressive complex 2 (PRC2) epigenetically regulates several biological processes, including cancer progression [109]. In such a context, the catalytic component of PRC2, the methyltransferase enhancer of Zeste homolog 2 (EZH2), another HSP90 client protein, is upregulated in several tumors, including breast and prostate cancers, and its overexpression correlates with a poor prognosis [110,111].

In the scenario of epigenetic mechanisms, DNA methylation fulfills a central role. DNA methyltransferases (DNMTs) are the writers of epigenome, and DNMTs have a role in the silencing of tumor-suppressor genes in cancer cells [112]. DNMT1 is the most abundant DNMT in adult cells [113] and is a target of HSP90 [90]. Interestingly, PcG cooperates with DNA methylation to regulate gene expression. Specifically, EZH2 employs DNMTs to target genes, and, on the other hand, DNMT activity targets PcG-marked genes. Given that EZH2 and DNMT1 are both targets of HSP90, it is not clear whether HSP90 interacts separately with these two proteins or acts as a scaffold to allow their cooperation [90].

A major function of HSP90 is to chaperone client proteins as kinases and transcription factors (TFs). In the class of TFs modulated by HSP90, there are, for instance, nuclear factor kB (NF-kB) and signal transducer and activator of transcription 3 (STAT3). Both NF-kB and STAT3 bind sites within the EZH2 promoter to upregulate its transcription. Furthermore, among the supposed mechanisms for “chromatinized” HSP90, there is also the interaction with the c-Myc oncogene, a driver of neoplastic transformation implicated in the uncontrolled proliferation and deregulated metabolism [114]. c-Myc has been reported to be a client of HSP90 [115,116], and is upregulated by HSP90 at a transcriptional level [99].

In addition to the well-known intracellular function, eHSP90 is also linked to epigenetic events. For example, eHSP90 influences the activation of epithelial-to-mesenchymal transition (EMT) in prostate cancer cells by modulating the expression and activity of EZH2. In this contest, eHSP90 activates MEK/ERK signaling, which is implicated in EZH2 transcriptional upregulation, and, consequently, induces tumor growth and invasion [117]. Furthermore, eHSP90 stimulates EGFR signaling [118] and elicits the activation of NF-kB [119] and NF-kB-STAT3 axes in cancer-associated stromal cells [120]. However, the link between eHSP90 and epigenetic regulation is not yet fully clarified, so additional investigations will be necessary in order to further understand this cellular phenomenon.

### 4.3. HSP90 Role in Metabolism–Epigenetics Crosstalk

Cancer cells are able to reprogram their metabolism so as to acquire and maintain malignant properties in order to adapt rapidly and survive in ever-changing environments [52,56,121,122]. This type of response requires a circuit in which the cellular metabolism and gene expression regulation must be bidirectionally connected and tightly coordinated. However, the influence of the metabolism on gene expression might involve epigenetic mechanisms, but this issue is still not well understood to date, and it is still a matter of substantial research because of its complexity.

The chromatin remodeling is a dynamic transcriptional mechanism highly responsive to external stimuli and environmental changes, such as nutrient availability [123]. Post-translational modifications of histones and DNA facilitate or prevent the recruitment of TF complexes, which will ultimately regulate gene expression. In such a context, many epigenetic enzymes require intermediary metabolites, such as cofactors or substrates, to remodel the chromatin structure. In addition, several metabolites are able to inhibit the activity of some epigenetic modifiers or to regulate their specificity. For this reason, each altered metabolic pathway may affect the epigenetic status of cells, generating gene expression patterns that sustain pathological conditions, like cancer [124,125,126,127].

The relationship between metabolism and epigenetics is not univocal, but a continuous crosstalk exists [5,121,122,128]. Indeed, as metabolic changes could alter the cellular epigenome, modifying the availability of epigenetic cofactors/substrates or influencing the activity of epigenetic effectors, consistently, a dysregulation of epigenetic pattern could influence the expression and/or the activity of metabolic genes, leading to metabolic reprogramming [5,129,130]. In such a context, HSP90s, acting as either drivers of metabolic rewiring or epigenetic capacitors, might have a role as linkers in the metabolism–epigenome crosstalk.

Recent research has started to reveal the unusual involvement of HSP90s in the epigenetic because of their crucial role in the modulation of cellular metabolic reprogramming. For example, as above mentioned, HSP90 directly binds to glycolytic enzymes, increasing the glycolytic flux and, consequently, the cellular availability of some glycolytic intermediate metabolites, among which is 3-phosphoglycerate (3PG), as demonstrated by Agarwal and colleagues [131]. 3PG can be shunted into the serine metabolism, for amino acid biosynthesis, and from here, into the folate/one carbon cycle fueling cells of precursors for the synthesis of the universal methyl donor SAM [132]. SAM is an essential co-substrate for the methyltransferase activity of DNMTs and HMTs, and an increase of its availability impacts profoundly on chromatin methylation, producing aberrant expression profiling. Generally, an excess of SAM is associated with the increment of global histone lysine methylation and DNA hypermethylation at CpG sites, and with the gene silencing of several key genes implicated in cancer progression and metastasis [133,134,135,136].

Oncogene-driven metabolic rewiring in cancer cells promotes glycolysis, thus supplying the cells of pyruvate with the synthesis of acetyl-CoA, the acetyl group donor for the acetylation of histones. In such a context, HSP90 chaperones favor Warburg metabolism in cancer cells, thus mimicking a condition of oncogene addiction [53]. These events, associated with conditions that favor the expression of acetyl-CoA synthase enzymes (i.e., ATP citrate-lyase and acetyl-CoA synthase short chain family member 1), lead to increased intracellular acetyl-CoA levels, and, in turn, histone acetylation, which is associated with the open chromatin and active expression [123] of the genes involved in cell cycle progression, proliferation, cell migration, metabolism, and macromolecular biosynthesis, upon remodeling of the chromatin structure [126,137,138].

In the last ten years, it is emerging that the metabolic reprogramming of cancer cells and mutations in metabolic enzymes, such as fumarate hydratase (FH), succinate dehydrogenase (SDH), and isocitrate dehydrogenase (IDH), cause the accumulation of particular intermediary metabolites with a driving role in cancer initiation and maintenance. The term “oncometabolites” indicates intermediates of metabolism, with oncogenic intracellular signaling function, which abnormally accumulate in cancer cells because of metabolic defects, often through loss-of-function or gain-of-function mutations of the genes encoding for the corresponding enzymes [139]. Succinate, fumarate, 2-hydroxyglutarate (both enantiomers *d* and *l*), and β-hydroxybutyrate are considered oncometabolites, because of their profound impact on the chromatin structure [140]. Intriguingly, the role of molecular chaperones in the production of oncometabolites is emerging. For example, TRAP1 contributes to the downregulation of mitochondrial respiration, inhibiting SDH [50,65,79], with an accumulation of succinate, and the consequent epigenetic alterations because of both TETs and JHDMs inhibition. In wild-type IDH1/2 tumors, under hypoxia or pseudohypoxia, like in a TRAP1 upregulated tumor environment [50], the production of the oncometabolite L-2-HG is frequent via the “promiscuous” reduction of α-KG, catalyzed by lactate dehydrogenase (LDHA) jointly with malate dehydrogenase (MDH) [141].

It is also worth noting the rising paradigm of gene regulation in which the subcellular compartmentalization, especially the nuclear localization, of specific metabolites or metabolic enzymes can modulate the expression of nearby genes [142]. The “local production and local consumption” model could explain the chromatin-localized biosynthesis of epigenetic intermediates, otherwise unstable or impermeable, to create a local “chromatin niche”, in order to direct the epigenetic modifications in specific chromatin regions [143]. The nuclear localization of HSP90s is reported, and beyond their role as epigenetic capacitors, another hypothetical chaperone function appears to be the protection of client proteins during their nuclear translocation or the modulation of their activity in the nucleus. For example, the embryonic isoform M2 (PKM2), a known interactor of HSP90 [64] resulting from an alternative splicing of PKM pre-mRNA, is a pivotal regulator of the Warburg effect [144], and is over-expressed in different types of cancer [145]. PKM2 nuclear translocation, under growth stimuli [146] or hypoxia [147], promotes cell proliferation and the reprogramming of the cancer metabolism, controlling epigenetic processes as well [144,146,148]. In the nucleus, it works synergistically with nuclear pyruvate dehydrogenase complex (PDC), providing acetyl-CoA locally for histone acetyltransferases activity [149]. In the yeast, the PKM2 homolog Pyk1 was found to be participating in the serine-responsive SAM-containing metabolic enzyme (SESAME) complex, involved in the upregulation of H3K4me3 [150]. Instead, GAPDH promotes histone acetylation, in response to cell stress, activating the GAPDH-mediated apoptotic pathway [151]. As previously mentioned, HSP90 binds to GAPDH [76], even though it is unclear if this direct interaction has a protective function on the GAPDH nuclear translocation or on its activity regulation.

In addition, some epigenetic enzymes have a notable role in the regulation of metabolic rewiring, and this may occur through the direct control of the HSP90s post-translational modifications and functions. The HDAC6-dependent deacetylation of cytosolic HSP90 is crucial for the stability of HSP90 interaction with several client proteins and co-chaperones [27,152,153]. Additionally, HSP90 hyper-acetylation decreases the binding to ATP, compromising chaperone functions [154] and suppressing multiple oncogenic signaling pathways, like those mediated by AKT [155,156]. Other histone deacetylases are involved in the tightly regulated process of dynamic HSP90 acetylation. Indeed, HDAC1 deacetylates HSP90 in the nucleus of human breast cancer cells and controls the expression and proteasomal degradation of DNMT1 [157]. Furthermore, both HDAC6 and HDAC10 were found to protect vascular endothelial growth factor VEGF receptor proteins 1 and 2 from proteasomal degradation upon the stabilization of VEGFR-HSP90 binding [158]. This evidence suggests that there is a bidirectional connection between epigenetic mechanisms and HSP90 chaperones, namely: that epigenetic enzymes regulate HSP90 chaperone activity, whereas altered HSP90 functions results in metabolic rewiring and modified epigenetic regulation.

Epigenetics might also indirectly regulate OXPHOS in cancer, controlling mitochondrial dysfunction through “anterograde regulation” (a mechanism of signaling from the nucleus to mitochondria), which promotes biogenesis and regulates mitochondrial activity to meet cellular needs. Recently, it was demonstrated how the mitochondrial metabolism might be impaired as a consequence of the interaction between the mitochondrial chaperone TRAP1 and SIRT3, a mitochondrial sirtuin. A positive feedback interplay was demonstrated, namely: that TRAP1 promotes the expression and stability of sirtuin, while SIRT3-mediated deacetylation regulates the chaperone’s activity. The interaction between TRAP1/SIRT3 fuels metabolic adaptation in the glioma stem cells, maintaining the stem-like phenotype [159].

Even though various and numerous modifications have been identified for their contribution to gene expression regulation by chromatin remodeling, much remains to be understood concerning the role of epigenetic–metabolism crosstalk in cancer initiation and progression. Furthermore, the participation of epigenetic effectors in HSP90s stability and how this might influence cancer cell metabolism is widely unexplored.

## 5. Conclusions: The Metabolism/Epigenetic Bidirectional Crosstalk as Novel Molecular Target

Molecular chaperones, such as HSP90, among others, act as hub proteins connecting a network of signaling pathways, because of their chaperoning activity on key cellular regulatory proteins, such as transcriptional factors, protein kinases, and hormone receptors. Thus, HSP90 chaperones are involved in essential cellular functions, including intracellular signaling, metabolism, and epigenetics, and are deregulated in many human diseases, such as neurodegenerative and metabolic diseases and malignancies. In such a perspective, particularly in cancer, HSP90 chaperones are regarded as potential targets for therapy, with the aim of reverting the molecular mechanisms of tumor progression. Hence, several inhibitors of HSP90 have been designed, and some of them entered in clinical trials [160]. Several preclinical and clinical studies suggest that HSP90 inhibition in cancer cells produces the suppression of cellular signaling, angiogenesis, metastasis, and survival. In pancreatic tumor models, the HSP90 inhibitors, NVP-AUY922 and ganetespib, were shown to suppress angiogenesis and tumor growth [161,162], whereas geldanamycin exhibited anticancer and pro-apoptotic activity in gastric cancer and cholangiocarcinoma cells, as s single agent or in combination with NVP-AUY922. Consistently, HSP90 inhibition by LD053 induced the inhibition of the pro-survival signaling pathway (i.e., c-RAF/MEK/ERK or PI3K/AKT) in gastric cancer cells [163]. However, in spite of this strong rationale, until now, HSP90 inhibitors failed to keep their promise in a clinical setting, and demonstrated several limitations either in terms of efficacy due to the incomplete inhibition of HSP90 [164], or in terms of toxicity. Nevertheless, the evidence that metabolic rewiring is a crucial and novel cancer hallmark [52,56], and that HSP90 chaperones have a central role in cellular adaptive pathways and in the crosstalk between epigenetic and metabolic pathways favoring metabolic rewiring, still support the strategy to inhibit HSP90 to target metabolism/epigenetic crosstalk. Consistently with this premise, the recent design of novel HSP90 inhibitors with better safety profiles [165] allows for the hypothesis that HSP90 inhibitors may be combined with anticancer therapies targeting epigenetic/metabolic mechanisms. In such a context, “epigenetic inhibitors” are under intense investigation and represent a promising therapeutic approach [166], because of the reversibility of the majority of epigenetic changes, and because they may provide a strategy to target cancer cell metabolism. DNMT inhibitors (i.e., 5-azacytidine and 5-aza-2′-deoxycytidine) were recently approved by the Food and Drug Administration (FDA) for the treatment of myelodysplastic syndromes and acute myeloid leukemia, respectively, based on their capacity to target DNA methylation induced by metabolic change. In the context of solid tumors, 5-azacytidine induced tumor regression in glioma xenograft models obtained from patients with the IDH1 mutation [167]. Recent evidence also suggests that HDAC inhibitors may affect cancer cell metabolism, namely: HT29 colorectal cancer cells treated with the HDAC inhibitors, butyrate, or trichostatin showed a reduction in lactate production and glucose uptake. Furthermore, multiple myeloma cells treated with valproic acid showed a significant reduction in GLUT1 expression and the inhibition of HKI activity [128].

Another area of intense investigation is to exploit cancer metabolism to revert DNA modifications, even though this field is in an early stage of development. Among the metabolic inhibitors targeting the epigenetic mechanisms, inhibitors of glycolysis and glutaminolysis or IDH1/2 inhibitors were investigated [128]. In cancer cells, glycolysis favors histone acetylation via citrate and acetyl-CoA, so its inhibition represents a therapeutic way to target histone acetylation. Glutaminolysis inhibitors may influence the epigenetic status, particularly in the contest of IDH1/2 mutations, by modulating the availability of the TCA cycle intermediates, such as acetyl-CoA and α-KG. AGI-5198, an IDH1 inhibitor, leads to histone demethylation, and causes a reduction in the production of the oncometabolite 2-HG [168].

As this represents an interesting and rapidly growing field of investigation, several issues still need to be addressed. Even though HDAC inhibitors have been approved only for hematological tumors, the potential benefit of these agents, in solid tumors as single agents or in combination with other drugs, still needs to be defined [169]. In this context, the combination with novel HSP90 inhibitors may represent a promising anticancer strategy to obtain synergistic effects [170]. Furthermore, preclinical/translational studies are needed in order to better investigate the metabolism–HSP90–epigenetics network in tumors, including biomarker analysis, in order to identify the epigenetic/metabolic markers to define the patient subsets that may take advantage of these synergistic therapeutic approaches, involving epigenetic regulators, metabolism inhibitors, and HSP90 pharmacological inhibition. 

## Figures and Tables

**Figure 1 cells-08-00532-f001:**
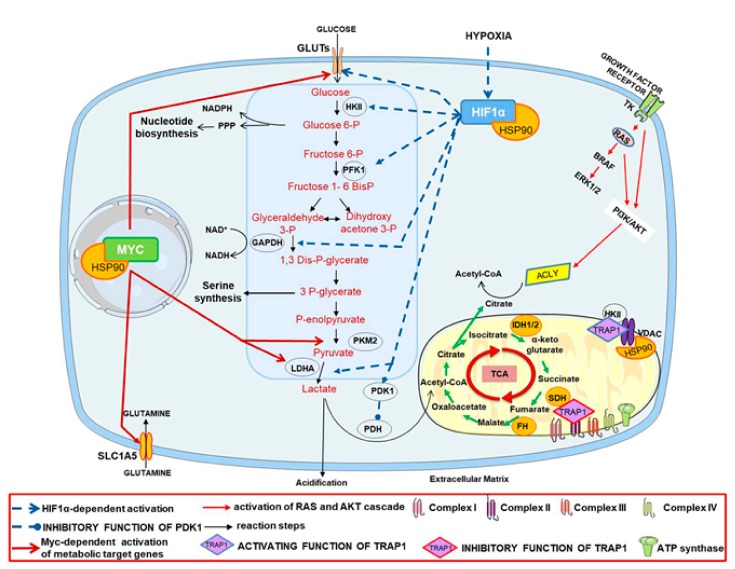
Heat shock protein 90 (HSP90) regulates metabolic pathways. HSP90s interact and modulate several cellular signaling pathways involved in the regulation of the metabolic key players in cancer cells. Indeed, HSP90 modulates the folding, stability, and activity of c-MYC that regulates the expression of the glucose transporter members (GLUTs), glutamine transporter (SLC2A5), lactate dehydrogenase (LDHA), and pyruvate kinase muscle isozyme 2 (PKM2), leading to increased glycolytic flow, the hyper-production of glycolytic intermediates used in biosynthetic pathways. In hypoxic conditions, HSP90 stabilizes hypoxia-inducible factor 1 alpha (HIF1α), which induces the expression of genes encoding for GLUTs and glycolytic enzymes, such as phosphofructokinase 1 (PFK1), glyceraldehyde 3-phosphate dehydrogenase (GAPDH), LDHA, and hexokinase II (HKII), and activates the transcription of the genes decreasing oxidative phosphorylation, such as pyruvate dehydrogenase kinase 1 (PDK1), an inhibitor of the tricarboxylic acid cycle. HSP90s interact with AKT, which activates ATP citrate lyase (ACLY), which controls the formation of acetyl-CoA from citrate. HSP90 mitochondrial isoform TRAP1 stabilizes the binding of HKII to the mitochondrial voltage-dependent anion channel (VDAC), maximizing its activity. TRAP1 also binds the respiratory chain complex II (succinate dehydrogenase—SDH), inhibiting its activity.

**Figure 2 cells-08-00532-f002:**
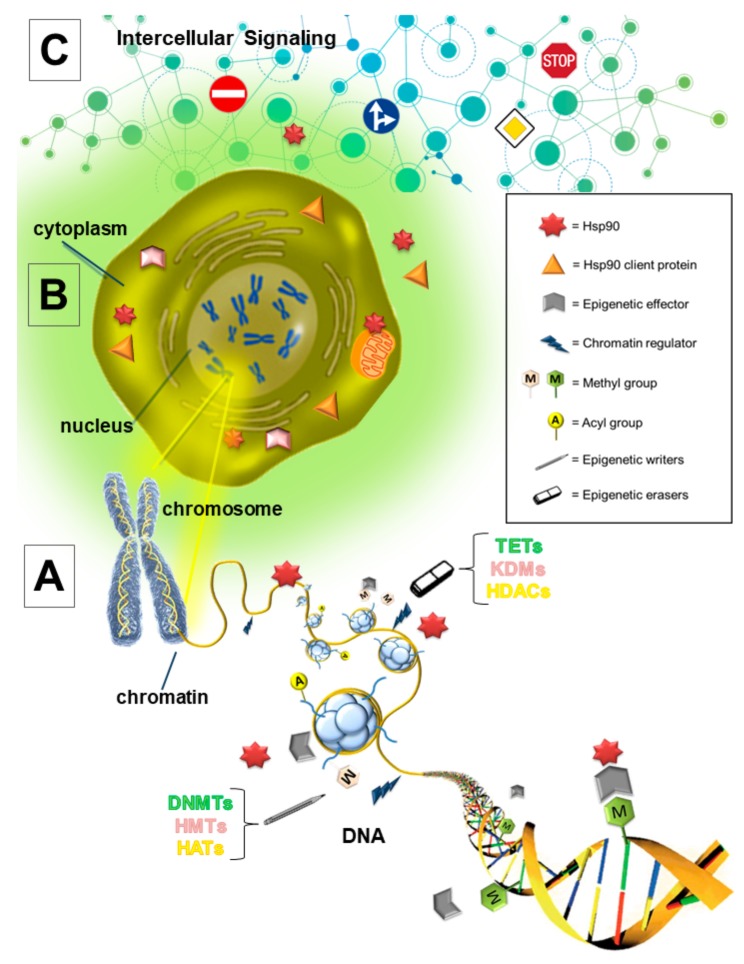
Hsp90 epigenetic mechanism of action. Hsp90 epigenetically regulates gene expression in a direct or indirect manner at an intracellular and/or extracellular level. (**A**) In the nucleus, Hsp90-chromatin interaction may be directly- or indirectly-mediated by the epigenetic effector or chromatin regulator. DNA methyltransferases (DNMTs) and ten eleven translocation hydroxylases (TETs) are epigenetic effectors that add or remove the methyl group on DNA, respectively. Hydroxymethyltransferases (HMTs) and lysine demethylases (KDMs) are responsible for histone methylation/demethylation, whereas histone acetyltransferases (HATs) and deacetylases (HDACs) are the competitors for the addition or removal of acetyl groups at histone lysine residues, respectively. (**B**) In the cytoplasm, the main function of Hsp90 is to chaperone the client proteins, such as kinases and transcription factors. Thus, in a contest-dependent manner, the Hsp90 client proteins play a role in regulating the epigenetic effectors. (**C**) In addition to the well-known intracellular function, the extracellular component of Hsp90 fulfills its function, particularly in the tumor microenvironment, and is linked to epigenetic events through the regulation of intercellular signaling.

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
