# Peer review of "HSP90 Molecular Chaperones, Metabolic Rewiring, and Epigenetics: Impact on Tumor Progression and Perspective for Anticancer Therapy"

_cells, 2019, doi:10.3390/cells8060532_

Round 1
Reviewer 1 Report
Condelli and co-workers prepared a manuscript about HSP90 chaperones connected to epigenetic and metabolism. Although this review is partly very detailed, it loses at the end the connection to the HSP90 chaperone machinery and ends up with a review about metabolism meets epigenetic in general. This point leads to the suggestion to rewrite the manuscript.
As explanation:
The manuscript starts with an introduction about heat-shock proteins, including a brief description of the HSP90 family. It continues with the role of HSP90 in cancer hallmarks, describing the hallmark ‘metabolism’ in more detail. In principle the lines 182-205 are not relevant for the further content, ‘metabolism and epigenetic’, of the manuscript. During the next part, the authors give some examples of HSP90 clients regulating metabolism. This part makes sense since it describes Hsp90 clients which are able to regulate metabolism. The fourth chapter is again very detailed and partly redundant, since it often repeats definitions and explanations of earlier parts, e.g. line 352-380 could be shortened. In principle, the following passages of this chapter describe now the connection between HSP90 and epigenetic which makes sense. But the last chapter, chapter 5, loses now completely the connection to the HSP90 system. It just describes all possible interplays between epigenetic and metabolism, very detailed, but without any clear explanations how HSP90 is involved. Within this complete chapter 5, I could not find any ‘HSP90’ wording anymore. That means I have to scroll back several times to find the Hsp90 clients which serve as bridge between epigenetic and metabolism. I would highly suggest to rewrite, or better somehow to include the chapter 5 into earlier chapters, to better explain the interplay/crosstalks between HSP90, epigenetic and metabolism. In this present form, chapter 5 feels like an own review, without access to HSP90.
Some minor concerns regarding chapter 1 - 4:
- Chapter 2 should explain the difference between the ‘normal’ and the ‘tumoral’ HSP90 system. The function of HSP90 is completely subverted during tumorigenesis, from a system which normally protects cells from aberrant proteins, to a system which now protects truncated, mutated and aberrant folded proteins. This is a major point to understand the nature of the tumoral HSP90 system to drive cancer. Further, tumoral HSP90 has now a higher affinity to their clients and built up the so-called super-chaperone complexes including HSP70, HSP40 and lots of co-chaperones. Such complexes need now much higher energy/ATP to fulfill protein protection/stabilization.
- It could be interesting to understand how nuclear HSP90 comes into the nucleus. Does it need transporters, etc.? HSP90 is known to protect proteins during the transport trough the cytosol to the nucleus, e.g. hormone receptors, but does not enter the nucleus. Normally, HSP90 complexes dissociate if a client is stabilized by their ligand. If this review focusses on epigenetic, the nuclear localization needs to be further investigated.
- It is known that the HSP90 activity is regulated by modifications such as acetylation. This manuscript mentioned HDAC regarding histone modification, of course. But what is with HSP90 activity?
- line 832: Ganetetispib
- line 853: the literature/citation is missing
Author Response
Reviewer #1
Condelli and co-workers prepared a manuscript about HSP90 chaperones connected to epigenetic and metabolism. Although this review is partly very detailed, it loses at the end the connection to the HSP90 chaperone machinery and ends up with a review about metabolism meets epigenetic in general. This point leads to the suggestion to rewrite the manuscript.
Comments | Authors Reply | ||
As explanation: The manuscript starts with an introduction about heat-shock proteins, including a brief description of the HSP90 family. It continues with the role of HSP90 in cancer hallmarks, describing the hallmark ‘metabolism’ in more detail. In principle the lines 182-205 are not relevant for the further content, ‘metabolism and epigenetic’, of the manuscript.
During the next part, the authors give some examples of HSP90 clients regulating metabolism. This part makes sense since it describes Hsp90 clients which are able to regulate metabolism.
The fourth chapter is again very detailed and partly redundant, since it often repeats definitions and explanations of earlier parts, e.g. line 352-380 could be shortened.
In principle, the following passages of this chapter describe now the connection between HSP90 and epigenetic which makes sense.
But the last chapter, chapter 5, loses now completely the connection to the HSP90 system. It just describes all possible interplays between epigenetic and metabolism, very detailed, but without any clear explanations how HSP90 is involved. Within this complete chapter 5, I could not find any ‘HSP90’ wording anymore. That means I have to scroll back several times to find the Hsp90 clients which serve as bridge between epigenetic and metabolism. I would highly suggest to rewrite, or better somehow to include the chapter 5 into earlier chapters, to better explain the interplay/crosstalks between HSP90, epigenetic and metabolism. In this present form, chapter 5 feels like an own review, without access to HSP90.
Some minor concerns regarding chapter 1 - 4:
- Chapter 2 should explain the difference between the ‘normal’ and the ‘tumoral’ HSP90 system. The function of HSP90 is completely subverted during tumorigenesis, from a system which normally protects cells from aberrant proteins, to a system which now protects truncated, mutated and aberrant folded proteins. This is a major point to understand the nature of the tumoral HSP90 system to drive cancer. Further, tumoral HSP90 has now a higher affinity to their clients and built up the so-called super-chaperone complexes including HSP70, HSP40 and lots of co-chaperones. Such complexes need now much higher energy/ATP to fulfill protein protection/stabilization.
- It could be interesting to understand how nuclear HSP90 comes into the nucleus. Does it need transporters, etc.? HSP90 is known to protect proteins during the transport trough the cytosol to the nucleus, e.g. hormone receptors, but does not enter the nucleus. Normally, HSP90 complexes dissociate if a client is stabilized by their ligand. If this review focusses on epigenetic, the nuclear localization needs to be further investigated.
- It is known that the HSP90 activity is regulated by modifications such as acetylation. This manuscript mentioned HDAC regarding histone modification, of course. But what is with HSP90 activity?
- line 832: Ganetetispib
- line 853: the literature/citation is missing
|
The description of cancer hallmarks was shortened, but kept to a minimum to help the understanding of subsequent paragraphs (lines 198-215).
According to reviewer suggestion, chapter 4 was shortened removing some redundant sentences (lines 385-406).
We agree with the Reviewer that chapter 5 loses the connection with HSP90. Thus, chapter 5 was amended to highlight the interplay between HSP90, metabolism and epigenetics. More specifically, we mentioned HSP90 client proteins involved in metabolic changes and potentially responsible for epigenetic modifications. However, it is important to note, that this is a field of intense investigation and that several issues are still unknown (Chapter 5, lines 561-563; 593; 628-629; 638-639; 688; 690-692; 757-758; 792; 823-824).
The concept that molecular chaperones play a different role in transformed versus non-transformed cells has been better specified. We underlined that cancer cells upregulate molecular chaperones to fold aberrant and mutated proteins and that they form heteroprotein complexes with client proteins and co-chaperones (Chapter 2, lines 123-127).
There is very limited information on the mechanism used by HSP90 to translocate to the nucleus. However, Chromatin IP-seq experiments showed that HSP90 directly interacts with chromatin. Furthermore, it has been proposed that HSP90 phosphorylation is a prerequisite for its nuclear translocation. The relevance of this mechanism in mammalian cells is still under investigation. These data were specified in the Review (lines 459-463).
As suggested by the Reviewer, the effect of post-translational modifications on HSP90 chaperones is a relevant issue. Thus, we reported some evidences about acetylation-dependent HSP90s activity. However, it is important to note that the effect of HSP90 deacetylation and/or acetylation on metabolism reprogramming and epigenetics is still unknown (lines 901-920; 926-931; 934-936).
The name of the inhibitor was corrected (line 948).
The reference was added (line 963). | ||

Reviewer 2 Report
The authors have presented a comprehensive review of the connections between HSP90 molecular chaperones and epigenetic signals in cancer. Overall, the manuscript is very well written and addresses the many points at which these seemingly unconnected topics overlap. This reviewer finds no significant complaints with the manuscript aside from the occasional stylistic concern (For example, Lines 400-402 are represented as a single paragraph-- a single sentence does not constitute a paragraph). Regardless, this does not diminish the quality of this review.
Author Response
Reviewer #2:
Comments | Authors Reply |
The authors have presented a comprehensive review of the connections between HSP90 molecular chaperones and epigenetic signals in cancer. Overall, the manuscript is very well written and addresses the many points at which these seemingly unconnected topics overlap. This reviewer finds no significant complaints with the manuscript aside from the occasional stylistic concern (For example, Lines 400-402 are represented as a single paragraph-- a single sentence does not constitute a paragraph). Regardless, this does not diminish the quality of this review. | Many tanks to the Reviewer for its favourable evaluation. Stylistic concerns were addressed. |
